# Vitamin D Deficiency in Testicular Cancer Survivors: A Systematic Review

**DOI:** 10.3390/ijms22105145

**Published:** 2021-05-13

**Authors:** Giuseppe Schepisi, Caterina Gianni, Sara Bleve, Silvia De Padova, Cecilia Menna, Cristian Lolli, Alessia Filograna, Vincenza Conteduca, Milena Urbini, Valentina Gallà, Chiara Casadei, Giovanni Rosti, Ugo De Giorgi

**Affiliations:** 1Department of Medical Oncology, IRCCS Istituto Romagnolo per lo Studio dei Tumori (IRST) “Dino Amadori”, Via P. Maroncelli 40, 47014 Meldola, Italy; caterina.gianni@irst.emr.it (C.G.); sara.bleve@irst.emr.it (S.B.); cecilia.menna@irst.emr.it (C.M.); cristian.lolli@irst.emr.it (C.L.); alessia.filograna@irst.emr.it (A.F.); vincenza.conteduca@irst.emr.it (V.C.); chiara.casadei@irst.emr.it (C.C.); giovanni.rosti@irst.emr.it (G.R.); ugo.degiorgi@irst.emr.it (U.D.G.); 2Psycho-Oncology Unit, IRCCS Istituto Romagnolo per lo Studio dei Tumori (IRST) “Dino Amadori”, Via P. Maroncelli 40, 47014 Meldola, Italy; silvia.depadova@irst.emr.it; 3Biosciences Laboratory, IRCCS Istituto Romagnolo per lo Studio dei Tumori (IRST) “Dino Amadori”, Via P. Maroncelli 40, 47014 Meldola, Italy; milena.urbini@irst.emr.it; 4Unit of Biostatistics and Clinical Trials, IRCCS Istituto Romagnolo per lo Studio dei Tumori (IRST) “Dino Amadori”, Via P. Maroncelli 40, 47014 Meldola, Italy; valentina.galla@irst.emr.it

**Keywords:** testicular cancer, vitamin D, deficiency, long-term, survivors

## Abstract

Testicular cancer (TC) is the most frequent tumor in young males. In the vast majority of cases, it is a curable disease; therefore, very often patients experience a long survival, also due to their young age at diagnosis. In the last decades, the role of the vitamin D deficiency related to orchiectomy has become an increasingly debated topic. Indeed, vitamin D is essential in bone metabolism and many other metabolic pathways, so its deficiency could lead to various metabolic disorders especially in long-term TC survivors. In our article, we report data from studies that evaluated the incidence of hypovitaminosis D in TC survivors compared with cohorts of healthy peers and we discuss molecular mechanisms and clinical implications.

## 1. Introduction

Testicular cancer (TC) is the most frequent tumor in young males [1]. In the vast majority of cases it is a curable disease; therefore, very often patients experience a long survival, also due to their young age at diagnosis [2]. Consequently, it is essential to consider all possible issues related to a long life expectancy, including second malignancies and cardiovascular, metabolic, and endocrinological diseases [3]. In the last decades, vitamin D deficiency after orchiectomy has become an increasingly debated topic [4]. Many studies have reported a correlation among hypovitaminosis D and different tumor types, but in patients with TC, given the young age at diagnosis, this deficiency could lead to long-term effects [5]. Indeed, vitamin D is essential in bone metabolism and many other metabolic pathways [6,7], so its deficiency could lead to various metabolic disorders, especially in long-term TC survivors. Vitamin D is mainly produced in the skin through exposure to sunlight, especially thanks to ultraviolet-B radiation (UVB, 290–320 nm). The precursor 7-dehydrocholesterol (7-DHC) present in the human skin is converted into an instable pre-vitamin D_3_ by a non-enzymatic process [8]. Subsequently a thermal isomerization by the sun’s heat quickly stabilizes it in cholecalciferol or vitamin D_3_ [9]. Some factors can modify the photoproduction of vitamin D, such as latitude, season, atmospheric conditions, clothing, sunscreen, and skin characteristics [10]. Furthermore nearly 20% of the vitamin D comes from dietary intake and includes ergocholecalciferol (vitamin D_2_), which is synthesized from ergosterol in vegetables after exposure to UVB radiation, and vitamin D_3_ from animal-derived foods. Oral intake is dependent on season, diet variation between countries, and intestinal absorption, and consequently, malabsorption syndromes can compromise the bioavailability of vitamin D, fats, and other liposoluble vitamins [11,12]. The two forms of vitamin D (D_2_ and D_3_) enter the bloodstream in a protein complex with the vitamin D binding protein (DBP) after absorption, and they are also stored in adipose tissue for subsequent release. The quantity of adipose tissue seems to inversely correlate with vitamin D status; it has been observed that obesity conduces to a decreased bioavailability of the vitamin [13].

The pre-hormone vitamin D is biologically inactive and requires a double activation in the liver and then in the kidney. Circulating vitamin D_3_ is converted to 25-hydroxyvitamin D_3_(25 (OH)D_3_ or calcidiol) by hydroxylation by the microsomal cytochrome P450 enzyme CYP2R1 (25-hydroxylase) in the liver. 25(OH)D_3_ is the main form of vitamin D measurable in blood and the principal indicator of vitamin D status [14]. 25(OH)D_3_ is then converted into the most active metabolite by hydroxylation by the mitochondrial enzyme 1α-hydroxylase (CYP27B1) producing 1,25(OH)_2_D_3_ (or calcitriol) and the isoform 24R,25(OH)_2_D_3_. This important activation takes place primarily in the proximal kidney tubule but also in cells of many other extra-renal sites, including immune system cells [15,16]. Calcitriol circulates binding DBP, but the high affinity for his respective nuclear receptor, called vitamin D receptor (VDR), determines its selective nuclear uptake from the bloodstream [17]. VDR heterodimerizes with the retinoid X receptor (RXR) interacting with specific DNA sequences, leading to the activation or inhibition of transcription and epigenetic effects. This nuclear receptor is expressed in many tissues, showing the wide range of effects of vitamin D in human metabolism [18]. Mutation and polymorphism in genes involved in vitamin D metabolism, catabolism, DBP, or VDR might be associated with altered vitamin D status and disease [19,20].

The main role of vitamin D is the control of calcium and phosphorus homeostasis, regulating calcium intestinal absorption, increasing the bone mineral content, and preventing bone loss and mineralization defects. Many in vitro and in vivo studies evaluated the involvement of 1,25(OH)_2_D_3_ and VDR in pleiotropic extraskeletal functions acting in many different transduction pathways in various sites of the body with genomic and non-genomic effects [21]. Vitamin D is implicated in the endocrine and immune systems, in the endothelial and cutaneous integrity, and also in fertility and reproduction [22,23,24,25]. An insufficient vitamin D status has been associated with higher cardiovascular risk, hypogonadism, and autoimmune disorders (psoriasis, inflammatory bowel disease, type I diabetes, multiple sclerosis, rheumatoid arthritis) [26,27,28].

Recently, the study of the relationship between vitamin D and cancer has been placed in the spotlight, starting from the investigation of its functions at the cellular level up to epidemiological research. 1,25(OH)_2_D_3_ has the potential to affect tumor growth, arresting the cell cycle progression in G1/S phase (action on cyclin-dependent kinase), promoting antiproliferative and proapoptotic pathways, initiating mechanisms of DNA damage repair, and enabling tumor attempts of invasion as epithelial-to-mesenchymal transition (EMT), angiogenesis, and metastasis [29,30,31]. Most convincing data from preclinical studies show how VDR can modulate the transcriptome in tumor cells, shaping micro RNA expression in a onco-protective way, and how VDR polymorphisms could increase cancer risk [32,33,34]. However many studies failed to find a real correlation between cancer risk and low calcitriol blood status [35]. Vitamin D daily supplementation has been shown to reduce overall cancer mortality by about 13% in 3–10 years of follow-up, whereas no statistical significance has been highlighted in reducing cancer incidence [36,37]. The protective effect of vitamin D_3_ intake also reduces the risk of high stage cancer of any type among adults without a diagnosis of cancer at baseline and normal body mass index [38]. More data from randomized controlled trials (RTCs) are expected to provide a better understanding of the impact of vitamin D in therapy, prevention, prognosis, risk of cancer, and follow-up outcomes in cancer survivors.

In our article, we report data from studies that evaluated the incidence of hypovitaminosis D in TC survivors compared with cohorts of healthy peers and we discuss molecular mechanisms and clinical implications.

## 2. Results

We found seven studies that evaluated vitamin D status in TC survivors. Data about vitamin D status from all the studies are summarized in Table 1, whereas correlations among vitamin D status and other parameters (e.g., testosterone, LH, FSH, etc.) are summarized in Table 2.

Firstly, we discuss two studies performed by the same authors. The first one was conducted in a small cohort of 15 TC survivors who underwent bilateral orchiectomy. At 3–5-year follow-up, they reported a significant difference in terms of vitamin D serum levels compared with a cohort of 41 healthy males (median serum levels 30.2 nmol/L versus 74.9 nmol/L, respectively). None presented nutritional deficiencies, and all of them received testosterone-replacement therapy, so their testosterone levels were into the normal range [39].

In 2013, Foresta et al. published a second study, conducted in a larger population of 125 TC survivors who underwent unilateral orchiectomy [40]. As a control group, the authors recruited the same cohort of 41 males from a center for sexual dysfunction, but all of them without testicular alterations. The authors tested not only vitamin D levels, but also several parameters involved in the mechanisms of osteopenia/osteoporosis and in vitamin D metabolism. A statistically significant difference between the two groups was found in terms of vitamin D serum levels: in fact, a value of 25OH-Vitamin lower than 50 nmol/L was reported in 77.6% of TC survivors and in 7.3% of control group (*p* < 0.0001). Similar differences were found in terms of bone mineral density (BMD) and incidence of osteopenia/osteoporosis according to the different follow-up period: in fact, bone disorders were reported in 23% of TC survivors and in 0% of control group, and their incidence was higher in patients with longer follow-up. Moreover, higher grades of osteopenia and osteoporosis were reported despite normal testosterone levels.

The potential biases of these two studies were the small sample size and the absence of a formal healthy male control group: the control group (41 subjects) was composed by other patients affected by other sexual dysfunctions. Moreover, the authors choose only a vitamin D cut-off level, but they failed to gather more information about it (e.g., insufficient versus deficient levels).

In the Netherlands, between 2007 and 2009, a small prospective study recruited 63 newly diagnosed TC patients, collecting blood samples at baseline and then every year for the first 5 years of follow-up. The authors divided patients into two cohorts: non-metastatic (patients who underwent unilateral orchidectomy only or who were also treated with adjuvant chemotherapy) versus metastatic patients. The authors reported a vitamin D deficiency (defined as 1,25(OH)_2_D_3_ serum levels <50 nmol/L) at baseline of 36.5%, which was equally distributed between the two study cohorts, and no significant modifications of these data were reported thereafter. In accordance with these findings, the metastatic cohort also demonstrated significant bone loss during the first year of follow-up, which was maintained even after 5 years [41]. The main biases of this Dutch study were the small number of recruited patients and those lost to follow-up.

In 2017, we conducted a study [42] testing vitamin D serum levels in a cohort of 61 long-term (3 or more years from diagnosis) TC survivors compared with another cohort of 40 healthy males of the same age. In our study, we found a significant (*p* = 0.031) difference in median vitamin D levels between the two groups (18.6 µg/L = 46.5 nmol/L in TC survivors and 23.6 µg/L = 59 nmol/L in healthy males, respectively). This difference was maintained during follow-up, even after 10 years from diagnosis, but at that time, the data were no more statistically significant.

The potential biases of our study were: (1) the small sample size (58 unilateral and 3 bilateral TC survivors); (2) the retrospective function of our analysis; thus, we collected serum specimens only after ≥3 years of follow-up—moreover, we were not able to collect blood samples before treatments, so the number of subjects who presented hypovitaminosis before tumor diagnosis is unknown; and (3) the period of the blood sample; in fact, all samples were collected only in winter, so we were not able to assess the seasonal variations in vitamin D levels according to amount of sun exposure.

Another Italian retrospective study evaluated variations in serum levels of sperm parameters and vitamin D in a cohort of 131 TC survivors who previously underwent pelvic radiotherapy. The authors also selected 61 TC patients treated only with simple surveillance as a control group. Blood samples were collected at baseline and at 1 and 2 years of follow-up. The authors reported a significant decrease of all sperm parameters at 1 year, whereas at 2 years these values were ameliorated. On the contrary, the LH levels progressively increased in the radiotherapy group, leading to a potential risk of subclinical hypogonadism. Moreover, those pretreated with radiotherapy showed a 5.78 relative risk of developing hypovitaminosis D (and consequently, osteopenia/osteoporosis), compared with the surveillance cohort [43]. The biases of the study were its retrospective nature and the absence of a direct measurement of bone status in the experimental cohort.

The sixth study was conducted in 82 TC patients. The authors reported reduced vitamin D levels in that patient cohort: in particular, 65–85% of them presented < 30 ng/mL (<75 nmol/L), 25–36% < 20 ng/mL (<50 nmol/L), and 6–18% < 10 ng/mL (<25 nmol/L). These results were observed at a median follow-up of 48 months, but the differences disappeared at a longer follow-up period [44]. The authors found no correlation between vitamin D deficiency and histology, stage, or antitumor treatment or with testosterone, FSH, or LH.

The limitations of this study were: (1) its retrospective nature, (2) small sample size (also a common limitation for the other studies), (3) lack of pre-orchiectomy collection of vitamin D serum levels, and (4) absence of a formal healthy male control cohort.

Recently, a German study was conducted in a cohort of 177 orchiectomized TC survivors with the aim of searching for a potential correlation between surgery and hypovitaminosis D. They collected samples before and immediately after surgery, and then at another 5 time-points until 2 years of follow-up. Their findings were compared with non-oncologic control cohorts, the first of them being a population of 84 males with testicular problems (Cohort 2) and the other one a group of 237 patients with urological diseases (Cohort 3), respectively. The comparison among the results from the study cohorts did not demonstrate any statistical difference in vitamin D levels (*p* = 0.161): the authors reported only a transient hypovitaminosis status starting from the postoperative period until 2 years of follow-up. After this period, the serum levels of vitamin D normalized again; therefore, the authors suggest that other organs may allow this late recovery, even if one testis is lacking [45]. The authors also tested other parameters, including testosterone, FSH, age, and histology, but no association among them and vitamin levels was found.

On a methodological level, the article by Dieckmann et al. [45] is more accurate than the previous ones: (1) it provided for a pre-orchiectomy dosage and immediately after surgery, with subsequent periodic samplings; (2) it had two control groups; (3) its sample size was much larger than in previous studies. The results therefore seem to be more supported also on a statistical level. However, the subdivision of patients into subgroups may have limited the sample size. Another potential bias was the seasonal variations in blood samples, which could affect the overall results.

## 3. Discussion

Currently, although an internationally validated optimal serum cut-off concentration of vitamin D has not been determined, 30 ng/mL (=75 nmol/L) and 20 ng/mL (50 nmol/L) are considered as upper cut-off levels for insufficiency and deficiency, respectively [7,46].

A small number of studies reported low vitamin D levels in TC patients compared with healthy matching controls [40,42,47,48]. However, as evidence of the uncertainty regarding the limit values, each study established its own cut-off levels: Foresta et al. chose a different cut-off to discriminate between normal levels and hypovitaminosis D (40 nmol/L in [49] and 50 nmol/L in [40]); Ghezzi et al. [42] and Willemse et al. [41] used an even higher cut-off level, >50 nmol/L, without distinguishing subjects with hypovitaminosis D; other authors [42,44] chose the abovementioned two cut-off levels, 30 ng/mL (=75 nmol/L) and 20 ng/mL (=50 nmol/L).

In the U.S., a study performed in a cohort of long-term survivors who underwent hematopoietic cell transplantation reported inadequate vitamin D levels in 35% of patients, even in cases of regular vitamin D supplementation [50]. These data could also be considered for TC survivors who underwent unilateral orchiectomy, as stem cell transplantation also represents a therapeutic option for them [51,52].

In our previous study [42], we also reported that lower levels were observed during the first year. Then, they remained stably low, even after ≥5 years of follow-up; the same variations were reported elsewhere [44]. In our study, we did not find any case of deficient vitamin D serum levels in the healthy male cohort, but we cannot exclude possible selection bias. In a supplemental study conducted within the French SUVIMAX project, the authors found 0–7% of hypovitaminosis D in a population of Mediterranean young males, unselected for TC [53]. Zitterman et al. found hypovitaminosis D in up to 40% of the general population in Germany [54]. In this regard, the article by Dieckmann et al. confirms the nadir of hypovitaminosis D at 6–12 months of follow-up; these values seem to be recovered after 2 years post-orchiectomy, but they are similar to the values found in the control groups [45]. Thus, including a healthy control group is mandatory in these studies, and its absence limits the results of the article, for example in Foresta et al. [40].

When assessing serum levels of vitamin D, its two main sources must first be considered: primarily solar exposure (especially UV radiation between 290 and 305 nm) and, secondly, diet: the first is responsible for the conversion into cholecalciferol (D_3_); the second is involved in the supply of ergocalciferol (D_2_) of plant origin. The sun exposure, of course, has seasonal variations; hence, vitamin D serum levels have differences as well, reaching their zenith at the end of summer [4]. Therefore, when testing serum levels of vitamin D, samples should be taken at different times of the year [27].

The efficiency of solar radiation in the formation of cholecalciferol is maximum at latitudes below 40°, while its effectiveness gradually decreases toward the poles, where diet can instead represent an excellent alternative source of vitamin D [11]. In this regard, the two German studies previously reported confirm a high rate of hypovitaminosis D in the German population [54] and in the two control groups [45], respectively. A study conducted in some nomadic populations of northern Russia, who commonly have a diet rich in ergocalciferol, has shown that hypovitaminosis occurs especially in those individuals who have abandoned traditional lifestyles to follow Western habits [55].

Several studies reported that lower vitamin D levels are frequently associated with hypogonadism [4,47,48], which is one of the long-term TC problems, not only after chemotherapy [56] but also after radiotherapy, as reported by Ghezzi et al. [43], probably through treatment-related damage to the Leydig cells—in fact, it is known that these cells are also involved in vitamin D 25-hydroxylation. In a study conducted in Slovakia, in a cohort of 823 unilateral TC survivors, the authors reported a 19.5% testosterone deficiency, a 19.1% LH increase, and 50.6% had osteopenia with/or osteoporosis [57]. More recently, the authors confirmed these results in an updated article on the same study, with more than 1200 subject enrolled [58].

In this regard, Sprauten et al. showed that the longer the follow-up, the higher the risk of hypogonadism. Moreover, they found an increased risk of precocious hormonal aging and age-related decline of sex hormones compared with healthy peers, mostly in pretreated TC survivors [59]. In this regard, some studies suggest a possible correlation with the expression of VDR both at the gonadal and hypothalamic level [4,47]; other authors highlight the common phylogeny among the members of the nuclear receptor family, which includes both VDR and sex hormone receptors [60]. Notwithstanding, a correlation among vitamin D, testosterone, LH, and FSH was not reported in some studies [42]. Dieckmann et al. tested these parameters, but they did not observe any correlation among them [45]. Higher serum levels of FSH have been described in TC survivors after surgery alone and especially after radiation/chemotherapy: a reduced fertility was reported in those patients [61].

However, even if the correlation between vitamin D and hypogonadism is not entirely clear, it must be considered that the expression of hydroxylases (CYP2R1 and CYP27B1) and VDR in testis tissue [47,62,63,64] also represents a further indication of the importance of the testicles in the metabolism of vitamin D [65,66,67]. It seems apparent in contrast with the finding of some vitamin D recovery starting from the second year of follow-up, but in reality, this could be explained in two ways: (1) progressive compensation by the contralateral testis (if unilateral), or (2) compensation by other organs that normally express hydroxylases CYP2R1 and CYP27B1, such as liver and kidney (especially in the case of bilateral tumor). This latter explanation seems to find confirmation in animal studies [27]. In contrast to those data, Willemse et al. demonstrated that bone loss persisted even after 5 years of follow-up [41].

As we previously said, VDR expression in testis could represent another clue in favor of the role of testes in the vitamin D metabolism. Nappi et al., having found different VDR expressions based on tumor histology (greater expression in non-seminomas than in seminomas), suggest a possible different expression even before surgery and even suspect some interaction with the oncogenesis mechanism. However, the same authors found no link between receptor expression and hypovitaminosis in their study cohort [44].

To date, in the International Guidelines on Survivorship [68], there are only generic recommendations about vitamin D supplementation in cases of deficiency, especially in long-term survivors; curiously, in Guidelines for TC patients this is not even reported [69,70,71].

According to Nappi et al. [44], whereas it has been demonstrated that stable insufficient vitamin D serum levels are related to a high risk of several pathologies, such as cardiovascular diseases [72,73], osteopenia/osteoporosis [40,74] and infertility [75], it would be useful to monitor this condition, especially in long-term CT survivors, so that the problem can be eventually remedied by adequate supplementation.

## 4. Materials and Methods

Our review was performed by following the PRISMA guidelines for reporting systematic reviews and meta-analyses [76] (Figure 1).

We conducted a systematic review of English-language literature until March 2021. Our research was performed into the main web databases (Medline, Scopus/EMBASE, and Google Scholar), with the aim of finding relevant studies dealing with the incidence of hypovitaminosis D in TC survivors. We used the following search terms: vitamin d OR cholecalciferol AND testicular OR testis OR germ cell AND cancer OR tumor. After reading the abstracts, we more thoroughly analyzed the articles’ full text. We also looked through their references in order to identify further interesting studies.

## 5. Conclusions

Analyzing the abovementioned studies, we found conflicting results in terms of vitamin D deficiency. The reasons are unclear, but the main study biases (in particular, small sample sizes and methodological issues) could in part represent an explanation for that. In the future, it would be appropriate to carry out prospective studies with larger sample sizes, and with adequately large control groups (including more patients with bilateral orchiectomy as well). Furthermore, on a methodological level, it would be essential to plan the blood samples at least at the baseline and annually, paying attention to any seasonal bias in the choice of the blood collection period.

## Figures and Tables

**Figure 1 ijms-22-05145-f001:**
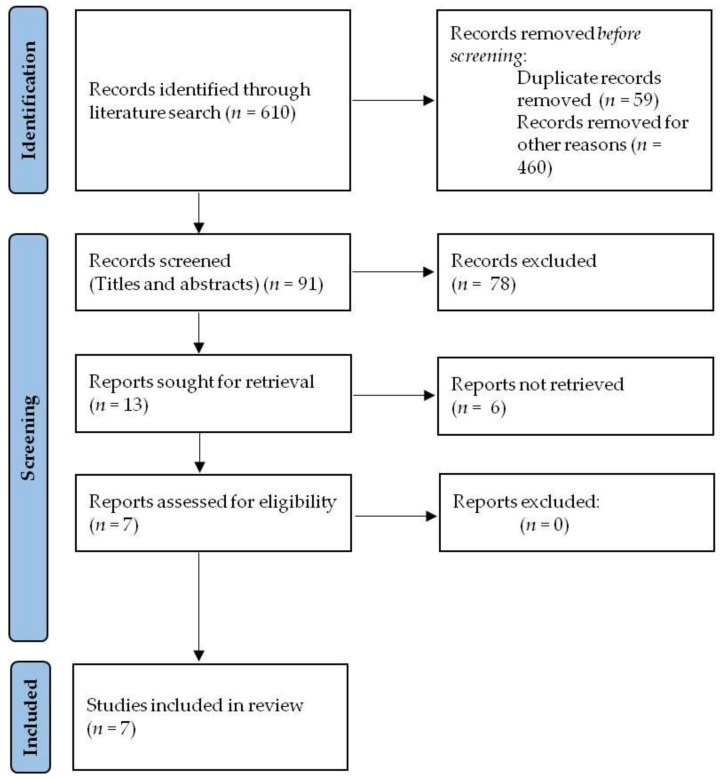
The search process according to the PRISMA Guidelines.

**Table 1 ijms-22-05145-t001:** Incidence of hypovitaminosis D among all studies. Abbreviations: Hypo-vitD: hypovitaminosis D; NA: not assessed; NR: not reported; RT: radiotherapy; T0: percentage of deficient vitamin D serum levels at baseline; T1: percentage of deficient vitamin D serum levels at 12 months; T2: percentage of deficient vitamin D serum levels at 24 months; T3: percentage of deficient vitamin D serum levels at 36 months; T4: percentage of deficient vitamin D serum levels at 48 months; T5: percentage of deficient vitamin D serum levels at 60 months and beyond; Tpre-s: percentage of deficient vitamin D serum levels before surgery; Tpos-s: percentage of deficient vitamin D serum levels immediately after surgery.

Study and Country	Patients	Hypo-Vit D Cut-Off Level	Hypo-Vit D in TC Survivors (%)	Hypo-Vit D in Healthy Control Group (%)	Time of Sample Collection
Foresta 2010 [39] ITA	15	<50 nmol/L	60%	NA	3–5 years
Foresta 2013 [40] ITA	125	<50 nmol/L	73.6%	7.3%	At baseline and at 3 months
Willemse 2014 [41] NED	63	<50 nmol/L	36.5%	NR	At baseline and then annually for 5 years
Schepisi 2017 [42] ITA	61	<75 nmol/L <50 nmol/L <25 nmol/L	81	0%	≥3 years
Ghezzi 2018 [43] ITA	192	<50 nmol/L	Survivors RT (T0) 27.9% 21.8% (T1) 32.8% 59.9% (T2) 47.6% 83.8%	NA	At baseline and then annually for 2 years
Nappi 2018 [44] ITA	82	<75 nmol/L <50 nmol/L <25 nmol/L	(T1) 85% (T2) 66% (T3) 80% (T4) 72% (T5) 81%	NA	At baseline, every 3 months for the first 2 years, then every six months until the fifth year
Dieckmann 2021 [45] DEU	177	<75 nmol/L <50 nmol/L <25 nmol/L	(Tpre-s) 78% (Tpos-s) 82% (T1) 97% (T2) 91% (T3) 77%	Cohort 2 79.8% Cohort 3 78%	Before and immediately after surgery, and then at 5 other time-points until 2 years of follow-up

**Table 2 ijms-22-05145-t002:** Synopsis of all parameters evaluated in the individual studies. Almost all studies demonstrated a statistically significant reduction in vitamin D values compared with controls. The variations of the other parameters with respect to controls are also reported. Abbreviations: DEU = Germany; ITA = Italy; NA = not assessed; NED = The Netherlands; NR = not reported; background is blue = no difference among TC survivors and controls; background is orange = higher levels in TC survivors than in controls; background is yellow = lower levels in TC survivors than in controls.

Study	25-OH VitaminD	Calcium	Phosphorus	PTH	Calcitonin	FSH	LH	Testosterone	Beta-Estradiol	Progesterone
Foresta 2010 [39]	*p* < 0.0001	NA	NA	NA	NA	NR		supplemented	NA	NA
Foresta 2013 [40]	*p* < 0.00001			*p* < 0.00001	NA	*p* < 0.00001	*p* < 0.00001			NA
Willemse 2014 [41]	(*p* = NR)	NR	NR	NR	NA					NA
Schepisi 2017 [42]	*p* = 0.047			*p* = 0.002		ns			*p* = 0.996	ns
Ghezzi 2018 [43]	*p*= 0.421				NA	*p* = 0.174			NA	NA
Nappi 2018 [44]	(*p* = NR)	NA	NA	NA	NA	(*p* = NR)			NA	NA
Dieckmann 2021 [45]	(*p* = 0.161)	NA	NA	NA	NA		NA		NA	NA

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
