# Peer review of "Vitamin D Deficiency in Testicular Cancer Survivors: A Systematic Review"

_ijms, 2021, doi:10.3390/ijms22105145_

Round 1

Reviewer 1 Report

Majority of patients of testicular cancer will experience a long survival for good disease control. Vitamin D is essential in many metabolic pathways and its deficiency could lead to various metabolic disorders. However, the role of the vitamin D deficiency related to orchiectomy is still under debate.

This review article investigated the incidence of hypovitaminosis D in testicular cancer survivors and the author also discussed molecular mechanisms and clinical implications of vitamin D.

In general, the manuscript is well written and the content provided comprehensive data and contributed to the field of vitamin D deficiency in testicular cancer survivors. I suggest this paper can be accepted for publication.

Author Response

RE: We sincerely thank the reviewer for this comment

Reviewer 2 Report

In this study, “Vitamin D deficiency in Testicular Cancer Survivors: A Systematic Review,” the authors investigated seven studies, including the most recent one, that have reported the vitamin D level after the testicular cancer treatments. The role of vitamin D is well known. Therefore, the authors focus on whether the orchiectomy leads to its deficiency and whether it would continue through the patient’s life. This article would facilitate the readers in the field. For more information and improved readability, please consider the following points.

Comments:

  1. It would be better if the authors also discuss why the vitamin D deficiency occurs after the orchiectomy. The reviewer wonders if the vitamin D deficiency is a typical or specific symptom of the orchiectomy. Does it occur after treatments for other cancers? 
  2. Are there any differences among types of or treatments for testicular cancers?
  3. How about summarizing the patients’ data in one table, such as size, age, treatments, season, and country in each report?
  4. Please use the same unit for the concentration of vitamin D. Alternatively, both the units, “µg/L” and “nmol/L,” are better to be shown together.
  5. The authors should explain whether the vitamin D receptor is expressed in the thalamus/hypothalamus with regard to the secretion of gonadotropins. 
  6. The sentence, “the role of the vitamin D deficiency related to orchiectomy has become an increasingly debated topic,” should be rewritten, such as “the vitamin D deficiency after orchiectomy has become an increasingly debated topic.” The original sentence is complex for the readers to understand the meaning. Some readers may recognize that the vitamin D deficiency has a good effect on the orchiectomy, or others may think that the vitamin D deficiency leads to the testicular cancer. Besides, it would be better if the authors explain why the vitamin D deficiency became a hot topic in the testicular cancer.

Author Response

In this study, “Vitamin D deficiency in Testicular Cancer Survivors: A Systematic Review,” the authors investigated seven studies, including the most recent one, that have reported the vitamin D level after the testicular cancer treatments. The role of vitamin D is well known. Therefore, the authors focus on whether the orchiectomy leads to its deficiency and whether it would continue through the patient’s life. This article would facilitate the readers in the field. For more information and improved readability, please consider the following points.

Comments:

  • It would be better if the authors also discuss why the vitamin D deficiency occurs after the orchiectomy. The reviewer wonders if the vitamin D deficiency is a typical or specific symptom of the orchiectomy. Does it occur after treatments for other cancers?

RE: We thank the reviewer for this suggestion: we added a specific sentence: “Many studies have reported a correlation among hypovitaminosis D and different tumor types, but in patients with TC, given the young age at diagnosis, this deficiency could lead to long-term effects”

  • Are there any differences among types of or treatments for testicular cancers? How about summarizing the patients’ data in one table, such as size, age, treatments, season, and country in each report?

RE: We thank the reviewer for this suggestion, but there are no data regarding variations for different therapies. It is the lack of one or both testicles that determines or worsens hypovitaminosis, as we specified in the discussion. As regards the size, it appears in table 1. We have added the country of each study in the same table.

  • Please use the same unit for the concentration of vitamin D. Alternatively, both the units, “µg/L” and “nmol/L,” are better to be shown together.

RE: We modified as requested, by inserting both the units in some specific cases.

  • The authors should explain whether the vitamin D receptor is expressed in the thalamus/hypothalamus with regard to the secretion of gonadotropins.

RE: we thank the reviewer for this interesting suggestion. In the Discussion section, we added a specific sentence (page 7 lines 9 to 13)

  • The sentence, “the role of the vitamin D deficiency related to orchiectomy has become an increasingly debated topic,” should be rewritten, such as “the vitamin D deficiency after orchiectomy has become an increasingly debated topic.” The original sentence is complex for the readers to understand the meaning. Some readers may recognize that the vitamin D deficiency has a good effect on the orchiectomy, or others may think that the vitamin D deficiency leads to the testicular cancer.

RE: we thank the reviewer for this suggestion; we modified that sentence as requested

  • Besides, it would be better if the authors explain why the vitamin D deficiency became a hot topic in the testicular cancer.

RE: We thank the reviewer for the suggestion: we added a specific sentence: “Many studies have reported a correlation among hypovitaminosis D and different tumor types, but in patients with TC, given the young age at diagnosis, this deficiency could lead to long-term effects”

Reviewer 3 Report

An interesting review evaluating the role of vitamin D deficiency in testicular cancer survivors. Althoguh the argument is controversial (not all the studies report similar associations) i found the paper interesting, and eligible to be published after minor revisions:

A Prisma flow chart for study selection would in my opinion be a great add to this study. also, Pubmed/MEDLINE may be considered a single database....have you searched also Google Scholar or Scopus/EMBASE? please modify accordingly.

Page 1 line 38-42  "Vitamin D is mainly produced in the skin through sunlight exposition, especially thanks to ultraviolet-B radiation (UVB, 290-320nm). The precursor 7-dehydrocholesterol (7-DHC) present in the human skin is converted into an instable pre-vitamin D3 by a non-enzymatic process" this paragraph needs a reference, such as: doi: 10.1007/s13668-020-00322-4.

Thank You

Author Response

An interesting review evaluating the role of vitamin D deficiency in testicular cancer survivors. Althoguh the argument is controversial (not all the studies report similar associations) I found the paper interesting, and eligible to be published after minor revisions:

  • A Prisma flow chart for study selection would in my opinion be a great add to this study. also, Pubmed/MEDLINE may be considered a single database....have you searched also Google Scholar or Scopus/EMBASE? please modify accordingly.

RE: We thank the review for this suggestion: we inserted a flow chart for study selection according to the 2020 PRISMA Guidelines

  • Page 1 line 38-42 "Vitamin D is mainly produced in the skin through sunlight exposition, especially thanks to ultraviolet-B radiation (UVB, 290-320nm). The precursor 7-dehydrocholesterol (7-DHC) present in the human skin is converted into an instable pre-vitamin D3 by a non-enzymatic process" this paragraph needs a reference, such as: doi: 10.1007/s13668-020-00322-4.

RE: we thank the reviewer for the suggestion, we added the reference accordingly